# A Flexible Sensor with Excellent Environmental Stability Using Well-Designed Encapsulation Structure

**DOI:** 10.3390/polym15102308

**Published:** 2023-05-15

**Authors:** Jian Zou, Zhuo Chen, Sheng-Ji Wang, Zi-Hao Liu, Yue-Jun Liu, Pei-Yong Feng, Xin Jing

**Affiliations:** 1Key Laboratory of Advanced Packaging Materials and Technology of Hunan Province, Hunan University of Technology, Zhuzhou 412007, Chinacz100520@163.com (Z.C.); lzh_132000@163.com (Z.-H.L.); yjliu_2005@126.com (Y.-J.L.); 2National and Local Joint Engineering Research Center for Advanced Packaging Material and Technology, Hunan University of Technology, Zhuzhou 412007, China; fpyedu@163.com

**Keywords:** hydrogel-based flexible sensors, stability, sensitivity, encapsulation, contact resistance

## Abstract

The hydrogel-based sensors suffer from poor stability and low sensitivity, severely limiting their further development. It is still “a black box” to understand the effect of the encapsulation as well as the electrode on the performance of the hydrogel-based sensors. To address these problems, we prepared an adhesive hydrogel that could robustly adhere to Ecoflex (adhesive strength is 4.7 kPa) as an encapsulation layer and proposed a rational encapsulation model that fully encapsulated the hydrogel within Ecoflex. Owing to the excellent barrier and resilience of Ecoflex, the encapsulated hydrogel-based sensor can still work normally after 30 days, displaying excellent long-term stability. In addition, we performed theoretical and simulation analyses on the contact state between the hydrogel and the electrode. It was surprising to find that the contact state significantly affects the sensitivity of the hydrogel sensors (the maximum difference in sensitivity was 333.6%), indicating that the reasonable design of the encapsulation and electrode are indispensable parts for fabricating successful hydrogel sensors. Therefore, we paved the way for a novel insight to optimize the properties of the hydrogel sensors, which is greatly favorable to developing hydrogel-based sensors to be applied in various fields.

## 1. Introduction

Flexible pressure/strain sensors with good flexibility have been developed and extensively studied due to their great potential in the application of wearable devices [1,2,3,4], soft robotics [5,6], artificial skin [7,8], and human-machine interfaces [9]. Conductive hydrogels are the most promising representative candidates for the fabrication of flexible sensors due to their excellent biocompatibility, adjustable mechanical properties, good self-healing properties, and similar mechanical compliance to a range of biological tissues from the brain to bone [10,11]. Long-term stability and high sensitivity are essential performance parameters when hydrogels are used as sensing materials. However, it is still a challenge to prepare hydrogels to balance these properties simultaneously.

Although water is an essential part of the hydrogels, it always suffers from significant loss along with the storage time, which further affects the hydrogel stability with the presence of mechanical hardening, low conductivity, high brittleness, and hard deformability [12,13,14]. To ensure the high environmental stability of the hydrogels, the most popular strategy is to encapsulate the prepared hydrogel conductor with Very High Bond (VHB) tape [15] or other elastic polymers such as Polydimethylsiloxane (PDMS) [16] and silicone [17]. However, perfect encapsulation is hard to achieve due to the poor adhesion of conventional hydrogel to the encapsulation layer. Recently, numerous efforts have been devoted to enhancing the hydrogel’s adhesion. For example, Gao et al. [18] fabricated an adhesive composite hydrogel based on hydrolyzed keratin protein-modified polyacrylamide that could adhere to different substrates through hydrogen bonding and metal complexation, and the maximum adhesion strength achieved was up to 450 N/m for the aluminum surface. A hydrogel with repeatable self-adhesion, in which the adhesion strength (8 kPa) remains almost constant for at least four cycles, was prepared by combining linear zwitterionic polymers with the physically crosslinked polyvinyl alcohol (PVA) network [19]. Moreover, inspired by marine mussels that can adhere to the surface of various substrates due to the presence of catechol in 3,4-dihydroxyphenyl-L-alanine (DOPA) and amine groups from lysine amino acids, polydopamine (PDA), which has excellent biocompatibility and a large amounts of catechol groups, has become a popular material to fabricate adhesive hydrogels [20,21,22,23,24,25]. Our group also used PDA to enhance the adhesive ability of the hydrogels [26,27]. For example, we prepared a multifunctional composite organohydrogel with high conductivity (36.7 mS/cm) and favorable adhesion strength (10.4 kPa) by oxidizing polymerized dopamine and partially reducing graphene oxide (GO) in a weakly alkaline environment.

Furthermore, flexible sensors always suffer from low sensitivity when employing hydrogel conductors. Thus, to address the problems, great efforts have been made and several solutions have been proposed, such as improving the conductivity of the hydrogel via coupling the ionic and electric conductive mechanisms [28]. For example, Zhang et al. [29] fabricated a multifunctional conductive PAM/CMCS-g-PANI/Ag hydrogel via a facile one-pot free radical polymerization. The hydrogel displayed high conductivity (0.45 mS/cm) and high sensitivity as a strain/pressure sensor (GF = 6.43) due to ionic/electronic dual conduction. However, the high conductive content of the fillers always deteriorates the performance of hydrogels. For example, salt ions can induce severe entanglement of polymer chains and form a denser network structure, which would further hinder the movement of ions, resulting in a reduction in conductivity [30,31]. In addition, serious entanglement of the polymer chains would increase the internal friction of the molecules, resulting in significant hysteresis. Therefore, the responsive signal of the sensors exhibits instability and large lag time as a result [32,33]. Moreover, Kim et al. [34] claimed that the material of the electrodes in the sensors also affects their sensitivity. Nowadays, metal and conductive tapes have been commonly used as electrodes to export signals [34,35]. Between the hydrogel and the electrode, the mobile ions/electrons from the hydrogel and the electrode form an electric double layer (EDL) at the interface [36], which functions as a capacitor to couple the currents from the hydrogel and the electrode, respectively, implying that the connecting state between the metal and hydrogels also plays a vital role in regulating the final signal visualized by the measuring system. In our lab, we also performed preliminary experiments to verify this hypothesis, and the results were highly consistent with that. However, there are very few studies to dig into the phenomena.

Herein, we prepared a Polyacrylamide/ sodium alginate/ Polydopamine/ polyaniline (PAM/SA/PDA/PANI) conductive hydrogel through “one-pot” polymerization, in which PDA endows the hydrogel with preferable adhesion but also facilitates the dispersion of PANI to enhance the conductivity of the hydrogel. Moreover, to enhance the long-term stability of the hydrogel sensors, a “special sandwich” structure was designed in which the conductive hydrogel was entirely encapsulated in Ecoflex with the help of a 3D-printed mold. Owing to the outstanding barrier performance of the Ecoflex and the robust interface binding force between the Ecoflex and hydrogel, the hydrogel-based sensors displayed superior environmental stability. In addition, the sensing performance of the sensor was quite stable even in 500-cycle tests. More importantly, we performed detailed theoretical analysis combined with simulation on the interface between the metal electrode and hydrogel. It was found that the sensitivity of the sensor can be adjusted by the contact state between the metal electrode and the hydrogel. Overall, we not only provide an innovative solution to the problem that hydrogel-based sensors are prone to failure due to water loss, but also propose a method to improve the sensitivity of the sensor from the perspective of electrodes, which is greatly favorable for the further development of hydrogel-based sensors.

## 2. Experimental

### 2.1. Materials

Acrylamide (AM, 99%), sodium alginate (SA, AR), ammonium persulfate (APS, 98%), N, N′-methylenebis-acrylamide (MBAA, 99%), dopamine hydrochloride (DA, 98%), 2,2′-azobis [2-(2-imidazolin-2-yl) propane] dihydrochloride (AIBI, >98%) and sodium hydroxide (NaOH, AR, 97%) were purchased from Aladdin Reagent Co., Ltd., China. Aniline (ANI, AR) was purchased from Shanghai Runjie Chemical Reagent Co., Ltd., Shanghai, China. Ecoflex was purchased from American smooth-on, American. Polylactic acid wire in 1.75 mm diameter was purchased from Shenzhen Chuangxiang 3D Technology Co., Ltd., Shenzhen, China. All chemicals were used as received unless further noted, and deionized water was used to prepare the hydrogels.

### 2.2. Preparation of Conductive Hydrogel

#### 2.2.1. Preparation of Polyaniline (PANI) Solution

First, a certain amount of ANI was injected into a 20 mL 2M HCl solution under stirring to obtain homogeneous solution A. Solution B was prepared by dissolving APS in a 20 mL 2M HCl solution. Then, solution B was added to solution A drop by drop under stirring for 12 h at 0 °C to obtain polyaniline (PANI) suspension liquid, and the PANI powder (nANI:nAPS = 1:1) was obtained after centrifugation and freeze-drying process.

#### 2.2.2. Preparation and Design Strategy of PAM/SA/PDA/PANI Hydrogel

First, 150 mg of SA was dissolved in 10 mL of deionized water under stirring for 1 h at 50 °C. Then, 6 mg DA was added to the above solution under stirring for 10 min and the pH of the mixed solution was adjusted to 8.5 via the prepared NaOH solution (1 mol·L^−1^). Subsequently, the solution was heated to 60 °C and stirred for 3 h to make the DA self-polymerize into PDA. Meanwhile, the as-prepared PANI powder was dispersed into deionized water by ultrasonication (JY92-IIDN Ultrasound Cell Breaker, Shanghai Jingxin Industrial Development Co., Ltd., Shanghai, China) for 30 min in an ice bath to obtain a 3 mg/mL PANI solution. Afterwards, 5 mL of the as-prepared PANI solution was added to the PDA solution. Furthermore, AM (3 g), MBAA (3 mg), and AIBI (60 mg) were added into the PDA/PANI solution step by step under stirring in an ice bath. Finally, the homogeneous solution was cast into a Petri dish for degassing and then placed in an oven at 60 °C for two hours to gelation completely. The cylindrical hydrogel was obtained by injecting the homogeneous solution into cylindrical molds with a height of 16 mm and an inner diameter of 15 mm, then placing them in a vacuum oven at 60 °C to crosslink for 2 h. In addition, PAM and PAM/SA hydrogel were also prepared following the same protocol for comparison.

The PAM/SA/PDA/PANI hydrogel was prepared via a facile “one-pot” polymerization that can be roughly divided into two parts, as shown in Figure 1. First, the DA monomer was dispersed in the SA solution, which formed a coating layer of PDA onto the SA molecular chains due to the self-polymerization of dopamine in the weak alkaline environment. Simultaneously, the color of the clear solution underwent an obvious change from light pink to brown in a few minutes and finally turned to deep brown-black over time, which was the typical color of DA polymerization. After that, the AM monomers and PANI solution were added to the above solution, in which the former polymerized to form a polyacrylamide network in the presence of initiator and crosslinker, and the latter acted as the conductive phase. In the prepared PAM/SA/PDA/PANI hydrogel, the SA chains and PAM network were linked to each other through interactions between the catechol groups of the PDA chains and the amino groups of the PAM network, the carboxyl group on the sodium alginate chain, respectively.

### 2.3. Assembly of Hydrogel Sensor

First, the Ecoflex solution was stirred until homogeneous, then degassed to eliminate air bubbles. The treated solution was poured into sleeve molds (as shown in Appendix A) and then placed in an oven at 60 °C for cross-linking and curing. The nickel-chromium wire in an Archimedes spiral shape that placed as an electrode on the thin Ecoflex layer in the sleeve molds. Then, the cylindrical hydrogel was placed in the center of the mold, and another nickel-chromium wire was placed on the upper surface of the hydrogel as another electrode. After that, the cover mold (as shown in Appendix A) was placed, and subsequently, a certain amount of Ecoflex solution was injected to submerge the hydrogel and then placed in an oven at 60 °C to cure the Ecoflex. Finally, the hydrogel-based sensor was obtained after peeling off the mold. Different hydrogel sensors with different-shaped nickel-chromium wires were also fabricated for comparison following the same step. Especially, for the straight-lined nickel-chromium wires as electrodes, they were slightly inserted into the upper and lower inner surfaces of the hydrogel to prevent tipping or floating when the Ecoflex was injected. A schematic diagram of the entire encapsulation process is shown in Figure 2.

### 2.4. General Characterizations

The Fourier Transform Infrared (FTIR) was used to evaluate the chemical structure of PANI, PAM hydrogel, and PAM/SA/PDA/PANI hydrogel. The scanning electron microscope (SEM) test was performed to display the microstructure of different samples, which can reflect the mechanical performance of different samples. The Fourier Transform Infrared Spectroscopy (FTIR) spectra of PANI, PAM hydrogel, and PAM/SA/PDA/PANI hydrogel were recorded using an FT-IR spectrometer (Bruker tensor 20, Brooke (Beijing) Technology Co., Ltd., Beijing, China) between 4000 and 400 cm^−1^ with a resolution of 4 cm^−1^. Before that, all the test samples were placed in an oven at 60 °C for 12 h to evaporate the water. The microstructure images of the pure PANI powder, PAM, and PAM/SA/PDA/PANI hydrogel were obtained using a scanning electron microscope (SEM, Tescan Mira3, Beijing Yiguang Technology Co., Ltd., Beijing, China). Before imaging, the hydrogel samples were fractured in liquid nitrogen and freeze-dried. Then, after gold sputtering, the samples were observed using an SEM at a high voltage of 3 kV.

### 2.5. Mechanical Tests

Hydrogel samples were shaped into a rectangle shape with a dimension of 30 mm × 10 mm × 2 mm (length × width× thickness) and clipped on the electronic universal testing machine (SAAS, Shenzhen Sansi Testing Technology Co., Ltd., Shenzhen, China) with a load of 100 N for the tensile properties. The tensile speed was set at 20 mm/min.

### 2.6. Adhesion Property Tests

The adhesion property of PAM/SA/PDA/PANI hydrogel was reflected by the adhesive strength that was performed on different substrates, including glass, steel, wood, porcine skin, and plastic. The lap-shear test is made of two substrates and a hydrogel to form the sandwich structure, as shown in Appendix A. In the lap-shear test, the contact area was kept as 10 mm × 20 mm between the substrates and PAM/SA/PDA/PANI hydrogel. Then, the substrates were stretched to failure using a speed of 10 mm/min on the electronic universal testing machine (SAAS) at room temperature.

### 2.7. Sensing Performance Test

An LCR meter (878B, B&K Precision Corporation, American) combined with the electronic universal testing machine was used to record the resistance signal during the compression process. Equation (1) was used to calculate the gauge factor (GF), which is an important value to evaluate the sensitivity of the sensors.
(1)$$GF=\left(\frac{R_{t}-R_{0}}{R_{0}}\right)/\varepsilon$$
where, *R_t_* and *R*_0_ are the real-time resistance and initial resistance, respectively. *ε* is the compressive strain, and *GF* represents the sensitivity of the sensor.

## 3. Results and Discussion

### 3.1. Characterization

To reveal and verify the interaction between the three components in the PAM/SA/PDA/PANI hydrogel, FTIR was used to characterize the synthetization of the prepared hydrogel (Figure 3a). As reported in the previous study, two characteristic peaks at 1473 cm^−1^ and 1561 cm^−1^ in the spectra of PANI were assigned to the stretching vibrations of the benzenoid and quinoid rings of PANI in the form of emeraldine salt, respectively [37]. The characteristic peak at 1298 cm^−1^ corresponded to the C−N stretching variation. In addition, the in-plane and out-of-plane bending of the C−H stretching of PANI were exhibited by the characteristic peaks at 1128 cm^−1^ and 805 cm^−1^, respectively [38]. In the FTIR spectra of PAM hydrogel, the characteristic peaks at 3350 cm^−1^ and 3178 cm^−1^ corresponded to the stretching vibration of the N–H bond, and the peaks at 1650 cm^−1^ and 1599 cm^−1^ assigned to the amide I band and amide II band of –CONH_2_, indicating the successful synthesis of PAM [27,39]. Compared with PANI powder and PAM hydrogel, the FTIR spectra of PAM/SA/PDA/PANI hydrogel displayed similar characteristic peaks but were shifted. For example, the peaks at 3350 cm^−1^ in PAM were shifted to 3355 cm^−1^and the peaks at 1128 cm^−1^ and 805 cm^−1^ in polyaniline were shifted to 1106 cm^−1^ and 816 cm^−1^. These might be because of the presence of hydrogen bonds between the PAM, PANI, SA, and PDA molecules. Moreover, a new peak at 1028 cm^−1^ was observed on PAM/SA/PDA/PANI hydrogel that corresponded to the C-O-C stretching vibrations in the pyranose ring, indicating the presence of the SA chain [40]. Another new peak at 1172 cm^−1^ was also observed on PAM/SA/PDA/PANI hydrogel that was assigned to C-O stretching vibrations, suggesting the presence of PDA [41,42].

A short rod-like polyaniline with uniform size was obtained by scanning electron microscopy (as shown in Figure 3b), indicating the successful preparation of polyaniline [43,44]. The morphology of the synthesized PAM and PAM/SA/PDA/PANI is shown in Figure 3c,d, respectively. It can be found that compared with the smooth pores observed in the PAM hydrogel, a larger number of pores with a rough surface are detected in the PAM/SA/PDA/PANI hydrogel, which might be because the strong interaction between the components in the hydrogel as well as the loaded PANI acted as nucleating agents in the formation of the pores.

### 3.2. Mechanical and Adhesive Properties

The representative stress-strain curves of the prepared PAM, PAM/SA, and PAM/SA/PDA/PANI hydrogels and their Young’s modulus are shown in Figure 4a,b, respectively. After introducing sodium alginate, the tensile strength and tensile modulus of the PAM/SA hydrogel reached 44 kPa and 27 kPa, respectively. However, for the PAM/SA/PDA/PANI hydrogel, it demonstrates the largest elongation at break with the lowest tensile strength, which might be because the presence of PDA affects the free radical polymerization of AM monomer during the fabrication process, which was consistent with previous studies [45,46].

Although the introduced PDA negatively affects the tensile strength of the hydrogel, the abundant catechol groups in the PDA endow the hydrogel with excellent adhesion properties which can interact with different surface through the formation of covalent and non-covalent bonds, such as hydrogen bonds and metal coordination bonds. The hydrogel can be stretched up to 800% without falling off when attached to fingers, implying its outstanding adhesion properties, as shown in Figure 4c. Then, lap-shear tests were conducted to quantitatively characterize the adhesion strength of the PAM/SA/PDA/PANI hydrogels to different surfaces. As shown in Figure 4d, the maximum adhesion strengths between the PAM/SA/PDA/PANI hydrogel and aluminum, rubber, plastic, wood, and glass were 11.5, 4.7, 1.4, 20.8, and 4.9 kPa, respectively.

### 3.3. Encapsulation Effect on the Sensor’s Stability and Response Performance

The environmental stability and response performance were used to evaluate the effect of encapsulation on the performance of the hydrogel-based flexible sensor. Figure 5a,b shows the change in weight and resistance of the encapsulated and unencapsulated hydrogel-based sensors after storing for one month in an indoor environment (temperature is about 25 °C, humidity is about 60%), respectively. The apparent changes in weight and resistance of the unencapsulated hydrogel-based sensor indicated that it was easily susceptible to ambient temperature and humidity. Moreover, the decrease in weight of the hydrogel-based sensor indicated the loss of water in the hydrogel, which further resulted in an increase in resistance. However, when the ambient temperature decreased and the humidity increased, the weight of the sensor increased because hydrogel absorbs moisture from the air, and the resistance showed the opposite trend. In contrast, it was found that after encapsulation, the relative changes in the weight and resistance of the sensor were only 1.9% and 7.7%, respectively, which were far less than those of 32.9% and 580.8% for the unencapsulated hydrogel-based sensor. In addition, a long-term cyclic step strain test was performed on the encapsulated hydrogel-based sensor to investigate its stability, and the results are shown in Appendix A. It can be seen that the relative resistance change measured in different periods has a high degree of coincidence, implying that the encapsulated hydrogel-based sensor has excellent long-term stability.

Unencapsulated hydrogel-based sensors exhibit a large hysteretic behavior during compression (Figure 5c) due to the inherent viscoelasticity of the hydrogel, while the hysteretic ring of encapsulated hydrogel-based sensors is significantly reduced (Figure 5d), indicating that Ecoflex can effectively improve the recovery performance of the sensors. The encapsulated hydrogel can return to its initial state after loading and unloading, as shown in Appendix A. The response performance of the hydrogel-based sensor before and after encapsulation was further tested. The relative resistance changes of the hydrogel-based sensor before encapsulation demonstrated a serious hysteresis phenomenon compared with its corresponding compression strain, and the lag time was 6 s (Figure 5e). While, for the encapsulated hydrogel-based sensor, owing to the excellent recovery performance of the Ecoflex layer, it can be seen from Figure 5f that the relative resistance curve was almost overlapped with its synchronous compression strain curve, demonstrating the outstanding responsiveness of the fabricated flexible sensor. Moreover, we also investigated the sensing ability of the sensor before and after encapsulation at different compression rates (10 mm/min, 50 mm/min, and 100 mm/min), and the results are shown in Appendix A, respectively. It can be found that the encapsulated hydrogel-based sensor exhibits significantly better stability under a fast compression rate, indicating that encapsulation effectively endows the hydrogel-based sensor with long-term stability and effectively improves its response ability.

### 3.4. The Influence of Material and Shape of Electrode on Sensor Performance

In addition to exploring the positive effects of encapsulation on hydrogel-based flexible sensors, the effects of the materials and shapes of electrodes used in the encapsulation process on the sensor’s performance were further investigated. Two different materials were selected as electrodes: conductive carbon tape and nickel-chromium wire. When the conductive carbon tape was used as the electrode of the hydrogel-based sensor, it can be seen that the output signals were unstable, which may be because the edge of the conductive carbon tape warps up during compression, causing it to separate from the hydrogel so that the resistance change is irregular (Appendix A). Moreover, the nickel-chromium wire was also chosen as a representative conductive electrode and verified for its feasibility in the hydrogel-based sensors. Even after storing for one month, the flexible sensor retained its original appearance (the height is 18 mm and the diameter is 20.5 mm) and could still respond to external stimuli. Moreover, 500 cycles of compression on the hydrogel-based sensor were also performed, which displayed excellent stability, as shown in Figure 6a. Furthermore, cyclic compression tests with step strains of 3%, 5%, 10%, and 20% were also conducted on the encapsulated hydrogel-based sensors, and the results are shown in Figure 6b. The resistance values upon compression and recovery process in each step strain tests were almost the same with a very slight deviation.

Moreover, electrodes with different contact states with hydrogels, including Archimedes spiral and linear electrodes, were used to investigate their effects on sensor sensitivity. As shown in Figure 6c, under the same strain, compared with the hydrogel-based sensor with the Archimedean spiral electrode, the relative resistance change rate of the hydrogel-based sensor with the linear electrode is smaller. In addition, the electrode with more Archimedean spiral cycles made a more significant contribution to the change in resistance of the hydrogel-based sensor during compression, resulting in a higher sensitivity. The gauge factor was calculated based on the relative resistance change of the hydrogel-based sensor, which is an important indicator of the sensitivity of the sensors. It can be seen that the GF of the sensor with the linear electrode slightly increased at first and then tends to be stable, which might be attributed to the fact that upon the small-strain compression, the conductive fillers in the hydrogel were gradually squeezed and then more conductive paths were formed, which reduced the resistance. When the compression was further performed, the conductive paths formed by the conductive components saturated, resulting in stable resistance. In contrast, the GF of the sample with the Archimedes spiral electrode first increased and then slightly decreased. The possible reasons might be as follows: Compared with the sensor with the linear electrode, in the initial compression stage, the sensor with the Archimedes spiral electrode not only changes in the number of conductive paths but also changes in the number of contact points between the electrode and the hydrogel, which is due to the uneven surface of the hydrogel (Appendix A). The synergistic effect makes the sensor resistance change significantly, resulting in a higher sensitivity. As long as the conductive channels were saturated and the connection between the hydrogel and the electrode stabilized, the change in amplitude of resistance was less significant. Therefore, a lower GF was present in the hydrogel sample, as shown in Figure 6d.

### 3.5. Theoretical Derivation and Simulation Analysis

Based on the above findings, it has been known that the change in contact state between the electrode and the hydrogel significantly affected the sensitivity of the hydrogel-based sensors. However, the mechanism behind that was still not explored. Therefore, in this study, the theoretical analysis and the simulation process were carried out to explore the detailed mechanism about that. Figure 7a,b displays the contact state schematic and the circuit schematic, respectively. It can be seen from the equivalent circuit diagram (Figure 7d) that the total resistance of the sensor was comprised of *R_n_* (the resistance of the electrode), *R_c_* (total contact resistance), and *R_h_* (resistance of the hydrogel). The reason for the generation of *R_sn_* (shrinking resistance of a single contact point) is that there are many contact points between the electrode and the hydrogel. When current flows through these contact points, the current lines shrink to produce *R_sn_*, as shown in Figure 7c [47,48,49]. It can be seen from Figure 6 that the number of *R_sn_* is not only related to the length of the electrode but also to the load applied to the electrode.

According to the classical Hertz theory, when a cylinder contacts an elastic plane, the relationship between the mechanical load *F* and an indentation depth *d* could be expressed by Equation (2) [50].
(2)$$F=\frac{\pi }{4}ELd$$
where, $E={\left[\frac{1-v_{1}^{2}}{E_{1}}+\frac{1-v_{2}^{2}}{E_{2}}\right]}^{-1}$ is a constant consisting of Young’s modulus and Poisson’s ratio of the involved two materials; *L* is the length of the cylinder, 2*a* represents the contact width and *r* is the electrode radius. Then, $a^{2}=r^{2}-{\left(r-d\right)}^{2}$, and because *d* is much smaller than *r*, it can be estimated as $a=\sqrt{\frac{rd}{2}}$. After substituting it into Equation (2), the relationship between the contact half-width *a* and *F* is obtained as shown in Equation (3).
(3)$$a=\sqrt{\frac{2rF}{\pi LE}}$$

The shrinking resistance of each contact point is expressed by Equation (4) [51].
(4)$$R_{sn}=\frac{2\rho }{\pi L}\ln\frac{2r+\sqrt{\left(a^{2}+{\left(2r\right)}^{2}\right)}}{a}$$
where, ρ is the contact resistivity. Normally, *a* is much smaller than 2*r*, so Equation (4) can be simplified into Equation (5):(5)$$R_{sn}=\frac{2\rho }{\pi L}\ln\frac{4r}{a}$$

When no external load is applied, *F* equals the contact pressure between the electrode and the hydrogel. According to Equation (3), the value *a* is not affected by *L*. Therefore, based on Equation (5), when the applied load is zero, $R_{sn}$ decreases with the increase in *L*. Substituting Equation (3) into Equation (5), the relationship between *F* and $R_{sn}$ can be obtained in Equation (6).
(6)$$R_{sn}=\frac{\rho }{\pi L}\ln\frac{8\pi rLE}{F}$$

Equation (6) shows that $R_{sn}$ is inversely proportional to *F*. As shown in the equivalent circuit of Figure 6, different shrinking resistances are parallel, and the *R_c_* can be expressed by Equation (7).
(7)$$\frac{1}{R_{c}}=\sum_{i=1}^{n}\frac{1}{R_{si}}$$

In the above equations, the units of force, length, and resistance are unified as Newton (N), millimeter (mm), and Ohm (Ω).

According to the law of parallel resistance, when the resistance on the branch decreases or the number of parallel resistances increases, the total resistance becomes smaller. Therefore, for a hydrogel-based sensor with Archimedean spiral electrodes, the increase in the number of contact points and the decrease in the resistance at each contact point during the compression process will lead to significant changes in the total contact resistance. In contrast, the relative resistance change for the sensor with linear electrodes is much smaller than that of an Archimedes spiral electrode under the same strain due to the fixed number of contact points. Moreover, based on the equations listed above, it can be known that when the hydrogel-based sensor was static, a longer electrode has greater number of contact points, and a lower value of *R_sn_* will generate a smaller total resistance. Therefore, for a hydrogel-based sensor with more Archimedes spiral-shaped loops, their smaller initial resistance would be present and the sensitivity would be higher.

Finally, the simulation analysis software Abaqus is used to verify whether classical Hertz theory is feasible in this system, and the simulation analysis model is provided in Appendix A. It can be seen from Figure 8a that in the longitudinal direction, the displacement gradually decreases from top to bottom; in the horizontal direction, the displacement decreases diffusely from the center to the periphery. In Figure 8b, the displacement cloud of the nickel-chromium wire electrode also reflects the same trend. When the strain is 20%, there is a deviation between the displacement of the hydrogel and the nickel-chromium wire electrode (Figure 8c). This shows that the nickel-chromium wire will be embedded in the hydrogel to a certain extent during compression, and the indentation depth (*d*) was used to indicate the degree of embedding. According to Equation (2) and the pressure applied on the nickel-chromium wire electrode obtained by the simulation analysis, the theoretical F-d curve could be obtained. Then the simulated *F*-*d* curve was obtained according to the simulation analysis value. As shown in Appendix A, the theoretical and simulated values have a relatively high coincidence, which indicates that the classical Hertz theory is applicable in our fabricated system.

## 4. Conclusions

In conclusion, we successfully fabricated an adhesive hydrogel by introducing PDA, which has a robust interface binding force (the adhesive strength reached 4.7 kPa), with silica gel. Furthermore, based on this hydrogel, we prepared a hydrogel-based sensor with a “special sandwich” structure utilizing a homemade 3D-printed mold. The encapsulated sensor exhibits excellent environmental stability with the help of the Ecoflex encapsulating layer, which displayed only 1.9% and 7.7% of weight and resistance changes when being directly exposed to the conventional environment for one month, respectively. Moreover, with the special encapsulated layer, the signal lag time of the sensor was decreased by 33% compared with the unencapsulated hydrogel-based sensor. In addition, the effect of the electrode on the performance of the hydrogel sensors was investigated theoretically and in simulation. It was found that the sensor’s sensitivity was closely related to the contact resistance between the hydrogel and the electrode. The sensitivity of the sensor was increased by 333.6% by optimizing the electrode. Through theoretical analysis and simulation verification, it is concluded that electrodes with more contact with the hydrogel surface can improve the sensitivity of the hydrogels, which proves that the feasible method proposed in this study might provide clearer guidance to adjust the sensitivity and stability of the hydrogel-based sensors in practical applications.

## Figures and Tables

**Figure 1 polymers-15-02308-f001:**
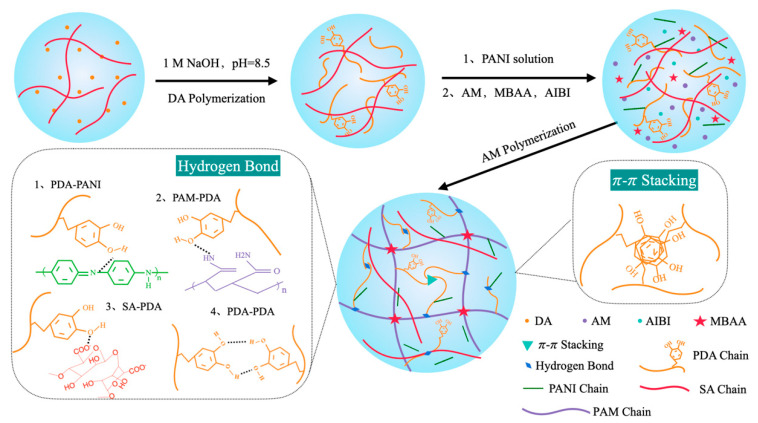
The schematic and fabrication process of the PAM/SA/PDA/PANI hydrogel.

**Figure 2 polymers-15-02308-f002:**
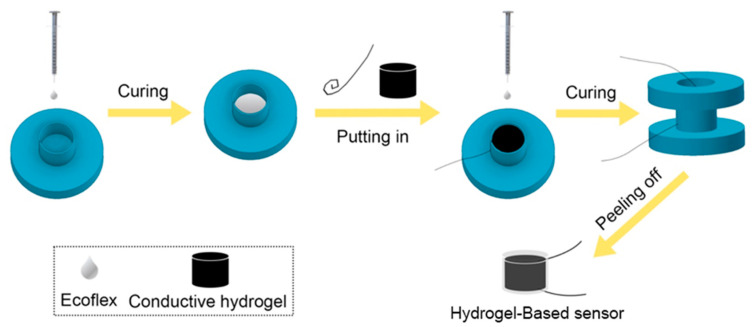
Schematic diagram of the encapsulation process using the printed mold.

**Figure 3 polymers-15-02308-f003:**
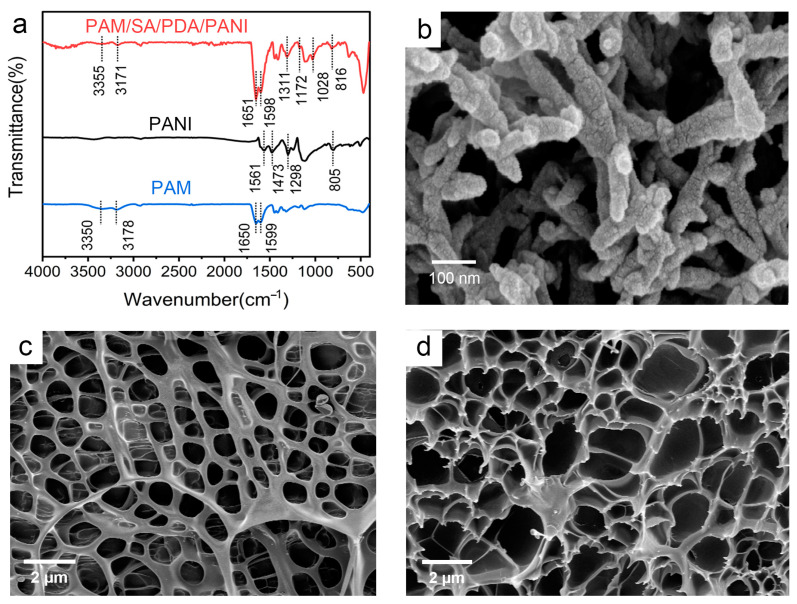
(**a**) FTIR spectra of PANI powder, PAM hydrogel and PAM/SA/PDA/PANI hydrogel; (**b**) SEM images of PANI powder; (**c**) SEM image of PAM hydrogel; (**d**) SEM image of PAM/SA/PDA/PANI hydrogel.

**Figure 4 polymers-15-02308-f004:**
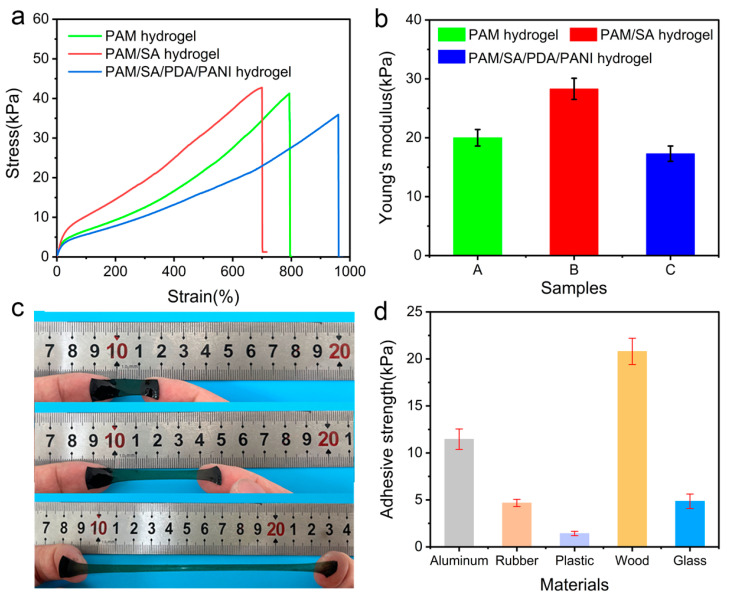
(**a**) The representative stress-strain curves of PAM, PAM/SA and PAM/SA/PDA/PANI; (**b**) The Young’s modulus of PAM, PAM/SA and PAM/SA/PDA/PANI; (**c**) The adhesion behavior of PAM/SA/PDA/PANI hydrogel upon stretching; (**d**) The adhesive strength of the PAM/SA/PDA/PANI hydrogel on difference surfaces.

**Figure 5 polymers-15-02308-f005:**
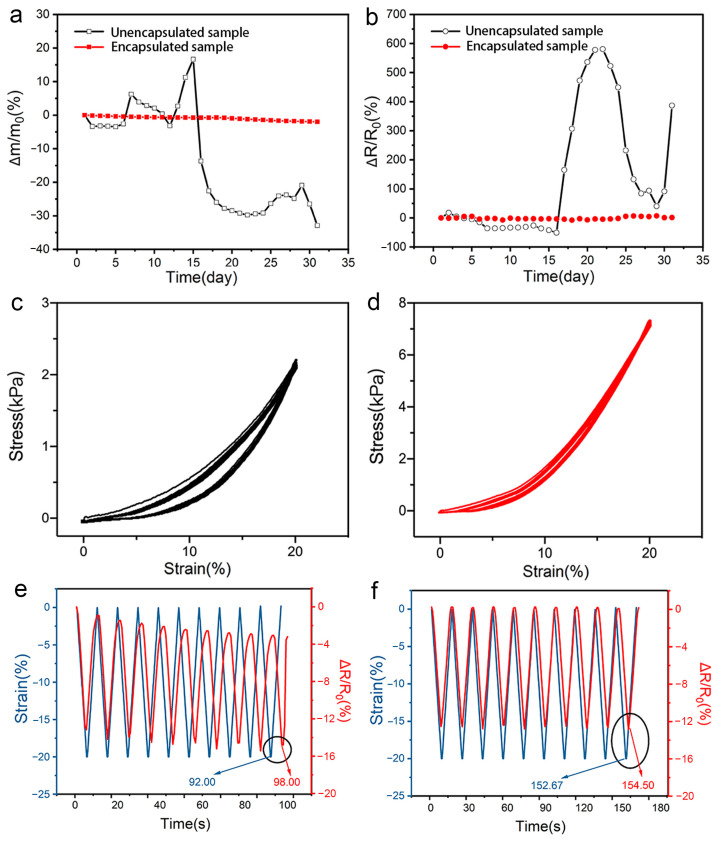
(**a**) The relative weight change of the hydrogel-based sensor before and after encapsulation; (**b**) The relative resistance change of the hydrogel-based sensor before and after encapsulation; (**c**) The cyclic compression curves of the hydrogel before encapsulation; (**d**) The cyclic compression curves of the hydrogel after encapsulation; (**e**) The lag behavior of the relative resistance change of the hydrogel-based sensor before encapsulation upon cyclic compression; (**f**) The lag behavior of the relative resistance change of the hydrogel-based sensor after encapsulation upon cyclic compression.

**Figure 6 polymers-15-02308-f006:**
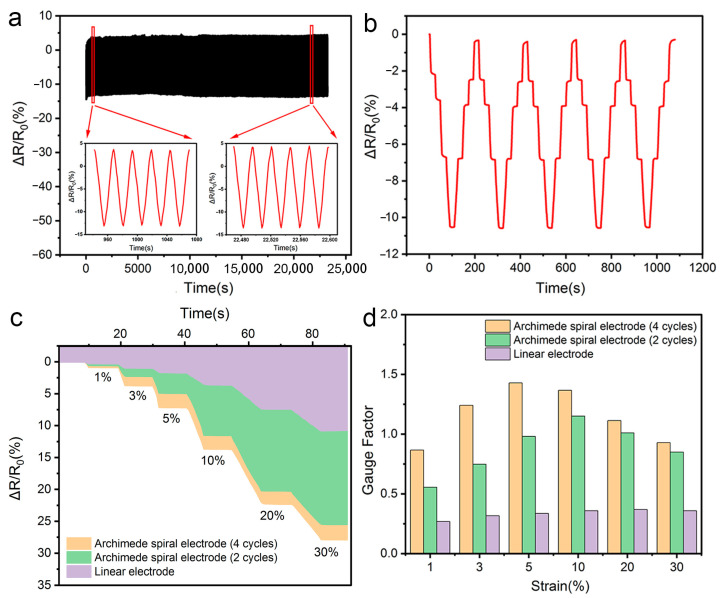
Testing results of the hydrogel-based sensor with nickel-chromium wire as electrodes. (**a**) The relative resistance change of the encapsulated hydrogel-based sensor upon 500 of cyclic compression tests; (**b**) The relative resistance change of the hydrogel-based sensor in the compression and recovery steps in the step strain tests; (**c**) The relative resistance change of the hydrogel-based sensor with different electrodes; (**d**) The GF of the hydrogel-based sensor with different electrodes upon compression.

**Figure 7 polymers-15-02308-f007:**
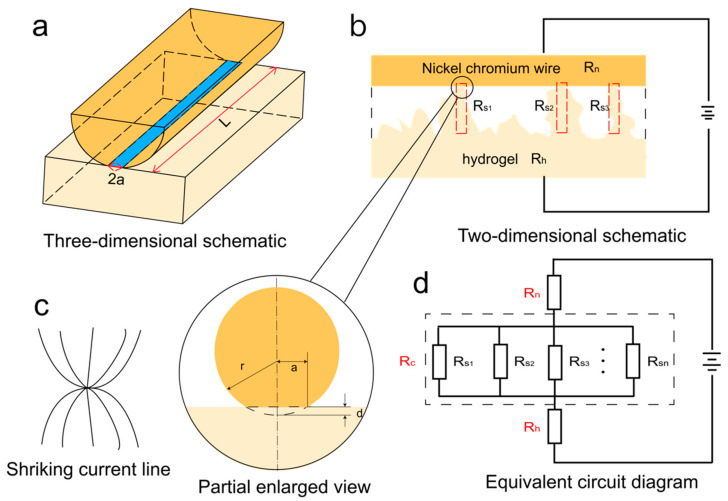
(**a**) Schematic of the contact state between the hydrogel and the metal electrode; (**b**) The circuit schematic; (**c**) Schematic diagram of shrinking current line; (**d**) The equivalent circuit diagram.

**Figure 8 polymers-15-02308-f008:**
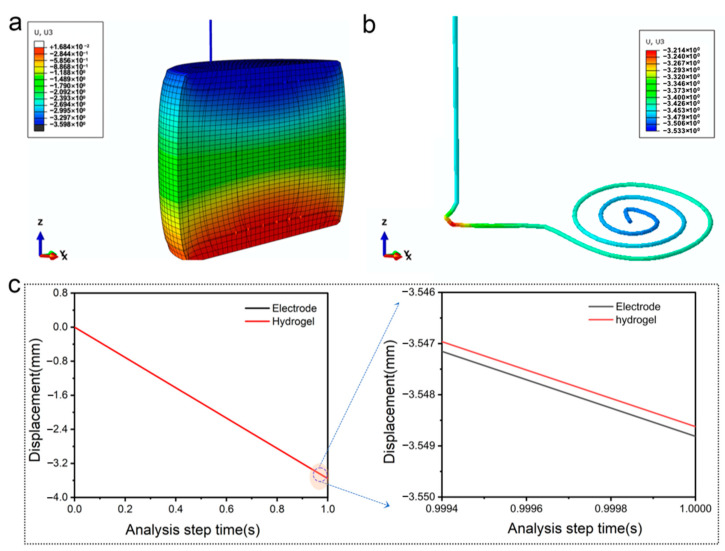
Simulation analysis results. (**a**) Displacement cloud diagram of sample; (**b**) Displacement cloud diagram of Nickel-chromium wire electrode; (**c**) Displacement-analysis time curve of hydrogel and Nickel-chromium wire electrode.

## Data Availability

The data is available upon reasonable request.

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
