# Peer review of "A Flexible Sensor with Excellent Environmental Stability Using Well-Designed Encapsulation Structure"

_polymers, 2023, doi:10.3390/polym15102308_

Round 1

Reviewer 1 Report

There is room for improvement for this manuscript. Comments are as follows.

Abstract - Key quantitative findings need to provide in the abstract. Besides, comparison of the developed hydrogen sensors with prior arts need to provide.

Figure 1 is nothing to do with the results. It should be presented in Methodology section.

Figure 2 - FTIR of PDA needs to be provided and discuss accordingly.

Figure 2(b) - Authors need to provide supporting references to prove the structure of the PANI.

Line 206 - Authors reported that the tensile strength and tensile modulus of the PAM/SA hydrogel were 44 kPa and 27 kPa, respectively. Are these values on par with other similar materials? Compare the data with other studies.

Section 3.3 and 3.4 - Authors should cite relevant references to compare with their findings.

Figure 5(a) - The dimension/size of the sensor need to be provided.

Equations (2)-(7) are part of the methodology. Why the authors present it in Results and Discussion section?

Figure 7(c) - The displacement between hydrogen and electrode (inset) is very small. Does the minor difference is so important?

Conclusion - Authors failed to provide key quantitative data in conclusion and how does the performance of the developed sensor compared to the state-of-the-art.

Author Response

Reply to the reviewer comments on the paper entitled “(polymers-2383668) A Flexible Sensor with Excellent Stability Using Well-designed Encapsulation Structure”

We are deeply grateful for the reviewer’ suggestions and comments on this paper. They have helped to improve the quality of our paper and the current research. We have made the following changes to accommodate their comments and suggestions in the revision.

Reviewers’ comments:

Reviewer #1:

There is room for improvement for this manuscript. Comments are as follows.

  1. Abstract - Key quantitative findings need to provide in the abstract. Besides, comparison of the developed hydrogen sensors with prior arts need to provide.

Reply to the comment:

Thanks for the suggestion. According to the comments, we have made the corresponding changes in the Abstract to indicate the quantitative findings in the revised manuscript as follows:

The hydrogel-based sensors suffer from poor stability and low sensitivity, severely limiting their further development. It is still “a black box” to understand the encapsulation as well as electrode on the performance of the hydrogel-based sensors. To address these problems, we prepared an adhesive hydrogel that could robustly adhere to Ecoflex (adhesive strength is 4.7 kPa) as encapsulation layer, and proposed a rational encapsulation model which fully encapsulated the hydrogel within in the Ecoflex. Owing to the excellent barrier and resilience of Ecoflex, the encapsulated hydrogel-based sensor can still work normally after 30 days, displaying excellent long-term stability. In addition, we performed theoretical and simulation analysis on the contact state between the hydrogel and the electrode. It was surprised to find that the contact state significantly affects the sensitivity of the hydrogel sensors (The maximum difference in sensitivity was 333.6%), indicating that the reasonable design of encapsulation and electrode are indispensable parts for fabricating successful hydrogel sensors. Therefore, we paved a novel insight to optimize the property of the hydrogel sensors, which is greatly favorable to developing hydrogel-based sensors to be applied for various fields.

  1. Figure 1 is nothing to do with the results. It should be presented in Methodology section.

Reply to the comment:

Thanks for the suggestion and the Figure 1 in the original manuscript has been placed in the Methodology section in the revised manuscript.

  1. Figure 2 - FTIR of PDA needs to be provided and discuss accordingly.

Reply to the comment:

Thanks for the suggestion. The characteristic spectrum of PDA was detected in the original FTIR results. According to the suggestion, we have added some relevant contents in the revised manuscript as follows:

Another new peak at 1172 cm-1 was also observed on PAM/SA/PDA/PANI hydrogel that was assigned to C-O stretching vibrations suggesting the presence of PDA[37, 38].

  1. Figure 2(b) - Authors need to provide supporting references to prove the structure of the PANI.

Reply to the comment:

Thanks for the suggestion. Due to the change of the layout of the figures in the revised manuscript, the structure of PANI was provided in the Figure 3. According to the suggestion, we have added some relevant references in the revised manuscript as follows:

Short rod-like polyaniline with uniform size was obtained by scanning electron microscope (as shown in Figure 3b), indicating the successful preparation of polyaniline[39, 40].

  1. Line 206 - Authors reported that the tensile strength and tensile modulus of the PAM/SA hydrogel were 44 kPa and 27 kPa, respectively. Are these values on par with other similar materials? Compare the data with other studies.

Reply to the comment:

Thanks for the suggestion. According to the suggestion, we found relevant references to compare the tensile strength and tensile modulus. For example, Sun prepared PAM/SA hydrogel with a tensile strength of 116 kPa and a tensile modulus of 16.9 kPa (Macromolecular Materials and Engineering, 2019, 304(10): 1900227, doi: 10.1002/mame.201900227). Zhang prepared PAM/SA hydrogel with a tensile strength of 85.4 kPa and a tensile modulus of 7.9 kPa (European Polymer Journal, 2023, 185: 111827, doi: 10.1016/j.eurpolymj.2023.111827). Luan prepared PAM/SA hydrogel with a tensile strength of 41.7 kPa and a tensile modulus of 6.5 kPa (Journal of Materials Chemistry C, 2022, 10: 7604-7613, doi: 10.1039/d2tc00679k). Although the mechanical properties of PAM/SA hydrogels in this work are somewhat different from those studies due to the different content of components, they are still within the acceptable range.

  1. Section 3.3 and 3.4 - Authors should cite relevant references to compare with their findings.

Reply to the comment:

Thanks for the valuable suggestion. In these two parts, we aimed to explore the effects of the encapsulation design on the long-term stability and response performance of the hydrogel-based sensor and the effects of electrode shape on sensor sensitivity, respectively, which was also the novelty of this work. However, there was no similar reports to indicate the findings in the recent studies.

  1. Figure 5(a) - The dimension/size of the sensor need to be provided.

Reply to the comment:

Thanks for the suggestion. According to the suggestion, we have added the dimensions of the sensor in the revised manuscript as follows:

Even after storing for one month, the flexible sensor retained its original appearance (the height is 18 mm and the diameter is 20.5 mm) and still could make response to external stimuli.

  1. Equations (2)-(7) are part of the methodology. Why the authors present it in Results and Discussion section?

Reply to the comment:

Thanks for the suggestion. In this part, we found that the shape of the electrode affected the sensor’s sensitivity during testing. In order to interpret the testing results clearer, we have quoted formulas here and carried out a series of derivations. It can be seen from the derived formula that the initial contact resistance and the change of the resistance during contact greatly influence the sensitivity of the sensor, corresponding to the experimental phenomenon.

  1. Figure 7(c) - The displacement between hydrogel and electrode (inset) is very small. Does the minor difference is so important?

Reply to the comment:

Thanks for the question. Due to the change of the layout of the figures in the revised manuscript, the results in Figure 7(c) was shown in Figure S9 in the revised manuscript. Due to the rough surface of the hydrogel, the displacement between the hydrogel and the electrode (inset) determines the number of contact points between them, which directly determines the value of the contact resistance, which further affects the sensitivity. The small displacement was caused by the small compression deformation (only 20%) of the sensor.

  1. Conclusion - Authors failed to provide key quantitative data in conclusion and how does the performance of the developed sensor compared to the state-of-the-art.

Reply to the comment:

Thanks for the suggestion. According to the suggestion, we have added some relevant key quantitative data in the revised manuscript as follows:

In conclusion, we successfully fabricated an adhesive hydrogel by introducing PDA, which has a robust interface binding force (the adhesive strength reached 4.7 kPa) with silica gel. Then based on this hydrogel, we prepared a hydrogel-based sensor with a "special sandwich" structure utilizing a homemade 3D-printed mold. The encapsulated sensor exhibits excellent environmental stability with the help of the Ecoflex encapsulating layer, which displayed only 1.9% and 7.7% of weight and resistance change when being directly exposed to the conventional environment for one month, respectively. Moreover, with the special encapsulated layer, the signal lag time of the sensor was decreased by 33% compared to the unencapsulated hydrogel-based sensor. In addition, the effect of the electrode on the performance of the hydrogel sensors was detailed investigated in theoretically and in simulation. It was found that the sensor’s sensitivity was closely related to the contact resistance between the hydrogel and the electrode. The sensitivity of the sensor was increased by 333.6% via optimizing the electrode. Through theoretical analysis and simulation verification, it is concluded that electrodes with more contact with the hydrogel surface can improve the sensitivity of the hydrogels, which proved that the feasible method proposed in this study might provide a clearer guidance to adjust the sensitivity and stability of the hydrogel-based sensors in practical applications.

Furthermore, the main research content of this manuscript is to study the effect of encapsulation on the performance of hydrogel-based sensor and the effect of electrode shape on sensor’s sensitivity. Therefore, we mainly focus on the improvement of sensor performance before and after encapsulation, and the relationship between electrode shape and sensitivity.

Reviewer 2 Report

This article proposed newly developed hydrogel-based sensors and tested a series of properties to validate corresponding functions. Overall, the results is interesting and will benefit the functionality and reliability of the hydrogel-based sensors. It can be published after addressing some minor issues:

1. It is recommended to promote the description of encapsulation structure from the supplemental information to the main content because in the title the encapsulation structure is included.

2. LIne 39, please spell out VHB.

3. Line 127, "The Fourier Transform Infrared..." Here it only said that which test was conducted, please also include the purpose of the test.

4. LIne 138-139. It says the load is 100N, test speed is 20mm/min. Is there any reason for these settings? What are the corresponding numbers in application conditions. 

5. Please add description of the lap-shear test.

6. Why there are highlighted parts in the manuscript?

7. Figure 3. Is the tensile modulus the Young's Modulus? If yes, please use Young's Modulus or elastic modulus because they are more commonly used.

8. LIne 230. Please spell out SI.

9. Line 232. Please describe conventional environment in more details.

10. Figure 4e&h. Some legends are missing. 

Author Response

Reviewer 2:

Comments and Suggestions for Authors

This article proposed newly developed hydrogel-based sensors and tested a series of properties to validate corresponding functions. Overall, the results is interesting and will benefit the functionality and reliability of the hydrogel-based sensors. It can be published after addressing some minor issues:

  1. It is recommended to promote the description of encapsulation structure from the supplemental information to the main content because in the title the encapsulation structure is included.

Reply to the comment:

Thanks for your constructive comments. We have already moved the description of encapsulation structure and figures into the manuscript as Section 2.3 in the revised manuscript as follows:

2.3 Assembly of hydrogel sensor

First, the Ecoflex solution was stirred until homogeneous, then degassed to eliminate air bubbles. The treated solution was poured into sleeve molds (as shown in Figure S1) and then placed in an oven at 60 °C for cross-linking and curing. The nickel-chromium wire in an Archimedes spiral shape that placed as an electrode on the thin Ecoflex layer in the sleeve molds. Then, the cylindrical hydrogel was placed in the center of the mold, and another same nickel-chromium wire was placed on the upper surface of the hydrogel as another electrode. After that, the cover mold (as shown in Figure S1) was placed, and subsequently a certain amount of Ecoflex solution was injected to submerge the hydrogel and then placed in an oven at 60 °C to cure the Ecoflex. Finally, the hydrogel-based sensor was obtained after peeling off the mold. Different hydrogel sensors with different shaped nickel-chromium wire were also fabricated for comparison following the same step. Specially, for the straight-lined nickel-chromium wires as electrodes, they were slightly inserted into the upper and lower inner surfaces of the hydrogel to prevent tipping or floating when the Ecoflex was injected. The schematic diagram of the entire encapsulation process is shown in Figure 2.

Figure 2. Schematic diagram of the encapsulation process using the printed mold.

  1. Line 39, please spell out VHB.

Reply to the comment:

Thank you for your comments. The full spelling of VHB is “Very High Bond”, We have added the full spelling of VHB in the revised manuscript where it first appeared, as follows:

To ensure the high environmental stability of the hydrogels, the most popular strategy is to encapsulate the prepared hydrogel conductor with Very High Bond (VHB) tape[12] or other elastic polymers such as Polydimethylsiloxane (PDMS)[13] and silicone[14].

  1. Line 127, "The Fourier Transform Infrared..." Here it only said that which test was conducted, please also include the purpose of the test.

Reply to the comment:

Thanks for your constructive comments. We have added the purposes of the test in section “2.3 General characterizations” in the revised manuscript as follows:

The Fourier Transform Infrared (FTIR) was used to evaluate the chemical structure of PANI, PAM hydrogel and PAM/SA/PDA/PANI hydrogel. The scanning electron microscope (SEM) test was performed to display the microstructure of different samples, which can reflect the mechanical performance of different sample.

  1. LIne 138-139. It says the load is 100N, test speed is 20mm/min. Is there any reason for these settings? What are the corresponding numbers in application conditions. 

Reply to the comment:

Thank you for the comment. We are truly sorry for the misunderstanding of a description of “a load of 100 N”. Actually, 100 N is the specification of the sensor of the electronic universal testing machine rather than the force applied to the hydrogel sensor. To avoid confusion, we have made corresponding changes in the revised manuscript as follows:

Hydrogel samples were shaped into a rectangle shape with a dimension of 30 mm × 10 mm × 2 mm (length × width× thickness) and clipped on the electronic universal testing machine (SAAS) with the load of 100 N for the tensile properties. The tensile speed was set as 20 mm/min.

The test speed was set as 20 mm/min, which was referred to the reported literature (ACS Appl. Mater. Interfaces, 2020, 12, 56509–56521, J. Mater. Chem. C, 2022,10, 11914-11923, Adv. Funct. Mater. 2023, 33, 2213895.).

  1. Please add description of the lap-shear test.

Reply to the comment:

Thank you for the comment. The lap-shear test is the common method to evaluate the adhesive performance in reported articles (Sci China Mater 2023, 66(1), 272–283, Mater. Horiz., 2020,7, 1872-1882, J. Mater. Chem. C, 2022, 10, 11914, Adv. Funct. Mater. 2023, 2302840, ACS Appl. Polym. Mater. 2021, 3, 54945508, J. Mater. Chem. C, 2022,10, 8266-8277 ).

For the section of “2.5 Adhesion property tests”, the description of the lap-shear test was added in the revised manuscript as follows:

The adhesion property of PAM/SA/PDA/PANI hydrogel was reflected by the adhesive strength that was performed on different substrates, including glass, steel, wood, porcine skin and plastic. The lap-shear test is made of two substrates and a hydrogel to form the sandwich structure, as shown in Figure S2. In the lap-shear test, the contact area was kept as 10 mm ´ 20 mm between the substrates and PAM/SA/PDA/PANI hydrogel. Then, the substrates were stretched to failure using a speed of 10 mm/min on the electronic universal testing machine (SAAS) at room temperature.”

Figure S2. Scheme illustration of the lap-shear test.

  1. Why there are highlighted parts in the manuscript?

Reply to the comment:

Thank you for the detailed comment. We are truly sorry for making the confusion. The highlights was caused by the editing before submitting and we forgot to remove that. In the revised manuscript, the highlights parts were already carefully checked and revised.

  1. Figure 3. Is the tensile modulus the Young's Modulus? If yes, please use Young's Modulus or elastic modulus because they are more commonly used.

Reply to the comment:

Thank you for the comment. Due to the layout change of the figures in the revised manuscript, the results in orginal figure 3 are shown in Figure 4 in the revised manuscript. For this test, the tensile modulus is Young’s modulus. We have replaced the tensile modulus with Young’s modulus in Figure 4b in the in the revised manuscript as follows:

Figure 4. (a) The representative stress-strain curves of PAM, PAM/SA and PAM/SA/PDA/PANI; (b) The Young’s modulus of PAM, PAM/SA and PAM/SA/PDA/PANI; (c) The adhesion behavior of PAM/SA/PDA/PANI hydrogel upon stretching; (d) The adhesive strength of the PAM/SA/PDA/PANI hydrogel on difference surfaces.

  1. Line 230. Please spell out SI.

Reply to the comment:

Thank you for your comments. The full spelling of SI is “Supporting Information”. We have added the full spelling of SI in the revised manuscript where it first appeared, as follows:

Finally, the simulation analysis software Abaqus is used to verify whether classical Hertz theory is feasible in this system, and the simulation analysis model was provided in Supporting Information (SI).

  1. Line 232. Please describe conventional environment in more details.

Reply to the comment:

Thank you for the comment. We are truly sorry for the misdirection of a description of “the conventional environment”. Actually, in this case, the conventional environment is just an indoor environment (temperature is about 25 °C, humidity is about 60%). (ACS Nano 2021, 15, 2698−2706; ACS Appl. Mater. Interfaces 2021, 13, 53055−53066; ACS Appl. Mater. Interfaces 2021, 13, 29008−29020; Adv. Mater. 2022, 2203650). Furthermore, we have replaced the conventional environment by indoor environment (temperature is about 25 °C, humidity is about 60%) in the revised manuscript as follows:

Figures 5a and 5b show the change of the weight and resistance of the encapsulated and unencapsulated hydrogel-based sensor after storing for one month under an indoor environment (temperature is about 25 °C, humidity is about 60%), respectively.

  1. Figure 4e&h. Some legends are missing.

Reply to the comment:

Thank you for the detailed comment. Due to the layout change of the figures in the revised manuscript, the results in orginal figure 4e&h are shown in Figure 5e&f in the revised manuscript. The legends have been added in the revised manuscript as follows:

Figure 5. (a) The relative weight change of the hydrogel-based sensor before and after encapsulation; (b) The relative resistance change of the hydrogel-based sensor before and after encapsulation; (c) The cyclic compression curves of the hydrogel before encapsulation; (d) The cyclic compression curves of the hydrogel after encapsulation; (e) The lag behavior of the relative resistance change of the hydrogel-based sensor before encapsulation upon cyclic compression; (f) The lag behavior of the relative resistance change of the hydrogel-based sensor after encapsulation upon cyclic compression

Reviewer 3 Report

Manuscript Number: polymers-2383668

Title: A Flexible Sensor with Excellent Stability Using Well-designed Encapsulation Structure

Article Type: Article

In the manuscript the experimental research concerning preparation of a flexible sensor composed of an electrode immersed in a novel polyacrylamide/sodium alginate/polydopamine/polyaniline conductive hydrogel encapsulated in Ecoflex is presented. As Authors indicate in the theoretical introduction flexible strain sensors are developed intensively nowadays due to their great potential in the application of wearable devices, soft robotics and many others. However the sensors based on the hydrogels suffer from many disadvantages such as poor stability or low sensitivity. Therefore, the authors set themselves the task of preparing a hydrogel that would solve the above-mentioned issues. Their goal was to perform a one pot reaction in order to prepare a hydrogel characterized by enhanced conductivity and enhanced long-term stability. Moreover some simulations were made using FEM analysis in order to investigate the properties of received material.

The hydrogels in sensing applications are investigated for decades. Yet, in my opinion the topic of the manuscript is relevant in the field. The wearable devices become more and more popular and therefore their improvement has fundamental importance. In the manuscript a simple procedure for synthesis of a hydrogel is presented which resulted in production of a material of satisfactory properties. The conclusions are consistent with the data presented in the manuscript and they do address the main research task posed.

In my opinion the biggest advantage of the manuscript is the development of a method to produce a novel hydrogel of enhanced properties and presentation of a broad range of measurements of mechanical properties of the material. The weakest point of the manuscript is the lack of comparison of received hydrogel properties with other data available in the scientific literature.

Below I am presenting my detailed remarks:

1.      The English of the manuscript should be enhanced – especially in the case of the abstract.

2.      Lines 188-190, 280 are marked in grey.

3.      Why does the stress in the stress-strain diagram is expressed in [%]? It should be expressed in pressure unit.

4.      The figures in the manuscript should be bigger. It is hard to read anything from them.

5.      Figure 5: What Does figure 5d present? What is the optical image? It is impossible to read the color scale in the corner of the picture.

6.      For the needs of numerical simulations the Poisson’s ratio had to be known. Please, present the data concerning this measurement in the manuscript.

7.      Please add the nomenclature with appropriate units for variables used in equations.

8.      The details concerning FEM simulation have to be presented.

9.      The FEM simulations should be compared with experiments.

10.   There are multiple mistakes in the References (e.g. “et al.”)

Multiple grammar mistakes.

Author Response

Reviewer #3:

In the manuscript the experimental research concerning preparation of a flexible sensor composed of an electrode immersed in a novel polyacrylamide/sodium alginate/polydopamine/polyaniline conductive hydrogel encapsulated in Ecoflex is presented. As Authors indicate in the theoretical introduction flexible strain sensors are developed intensively nowadays due to their great potential in the application of wearable devices, soft robotics and many others. However the sensors based on the hydrogels suffer from many disadvantages such as poor stability or low sensitivity. Therefore, the authors set themselves the task of preparing a hydrogel that would solve the above-mentioned issues. Their goal was to perform a one pot reaction in order to prepare a hydrogel characterized by enhanced conductivity and enhanced long-term stability. Moreover some simulations were made using FEM analysis in order to investigate the properties of received material.

The hydrogels in sensing applications are investigated for decades. Yet, in my opinion the topic of the manuscript is relevant in the field. The wearable devices become more and more popular and therefore their improvement has fundamental importance. In the manuscript a simple procedure for synthesis of a hydrogel is presented which resulted in production of a material of satisfactory properties. The conclusions are consistent with the data presented in the manuscript and they do address the main research task posed.

In my opinion the biggest advantage of the manuscript is the development of a method to produce a novel hydrogel of enhanced properties and presentation of a broad range of measurements of mechanical properties of the material. The weakest point of the manuscript is the lack of comparison of received hydrogel properties with other data available in the scientific literature.

  1. The English of the manuscript should be enhanced – especially in the case of the abstract.

Reply to the comment:

Thanks for the valuable suggestion. We have made corresponding improvements to grammar and spelling in the revised manuscript.

  1. Lines 188-190, 280 are marked in grey.

Reply to the comment:

Thank you for the detailed comment. We are truly sorry for making the confusion. The highlights were caused by the editing before submitting and we forgot to remove that. In the revised manuscript, the highlights parts were already carefully checked and revised.

  1. Why does the stress in the stress-strain diagram is expressed in [%]? It should be expressed in pressure unit.

Reply to the comment:

Thank you for the detailed comment. We are truly sorry for the mistakes. The unit of pressure has been revised into pressure as indicated in the Figure 4 in the revised manuscript as follows:

Figure 4. (a) The representative stress-strain curves of PAM, PAM/SA and PAM/SA/PDA/PANI; (b) The Young’s modulus of PAM, PAM/SA and PAM/SA/PDA/PANI; (c) The adhesion behavior of PAM/SA/PDA/PANI hydrogel upon stretching; (d) The adhesive strength of the PAM/SA/PDA/PANI hydrogel on difference surfaces.

  1. The figures in the manuscript should be bigger. It is hard to read anything from them.

Reply to the comment:

Thanks for your constructive comments. According to the suggestions, the layout of the figures has been revised to present clearer and some figures were placed into the supporting information file. Detailed revision was shown in the revised manuscript as follows:

Figure 5. (a) The relative weight change of the hydrogel-based sensor before and after encapsulation; (b) The relative resistance change of the hydrogel-based sensor before and after encapsulation; (c) The cyclic compression curves of the hydrogel before encapsulation; (d) The cyclic compression curves of the hydrogel after encapsulation; (e) The lag behavior of the relative resistance change of the hydrogel-based sensor before encapsulation upon cyclic compression; (f) The lag behavior of the relative resistance change of the hydrogel-based sensor after encapsulation upon cyclic compression

Figure 6. Testing results of the hydrogel-based sensor with nickel-chromium wire as electrodes. (a) The relative resistance change of the encapsulated hydrogel-based sensor upon 500 of cyclic compression tests; (b) The relative resistance change of the hydrogel-based sensor in the compression and recovery steps in the step strain tests; (c) The relative resistance change of the hydrogel-based sensor with different electrodes; (d) The GF of the hydrogel-based sensor with different electrodes upon compression

Figure 8. Simulation analysis results. (a) Displacement cloud diagram of sample; (b) Displacement cloud diagram of Nickel-chromium wire electrode; (c) Displacement-analysis time curve of hydrogel and Nickel-chromium wire electrode

  1. Figure 5: What Does figure 5d present? What is the optical image? It is impossible to read the color scale in the corner of the picture.

Reply to the comment:

Thanks for the suggestion. It should be noted that in order to make the picture easier to read, we have changed the layout of the Figure. The results in Figure 5d was shown in the Figure S7 in the revision. Figure S7 is a 3D depth-of-field image of PAM/SA/PDA/PANI hydrogel, which can prove that the surface of the hydrogel is rough. Furthermore, according to the suggestion, we have enlarged the color scale in the corner of the picture in the revised manuscript as follows:

Figure S7. The 3D depth-of-field image of cylindrical hydrogels

  1. For the needs of numerical simulations the Poisson’s ratio had to be known. Please, present the data concerning this measurement in the manuscript.

Reply to the comment:

Thanks for the valuable suggestion. We are truly sorry for the lack of relevant information in the manuscript. According to the suggestions, we have added the details of the Poisson’s ratio test in the revised Supporting information as follows:

The Poisson’s ratio of nickel chromium wire was obtained by querying the manufacturer. In order to obtain the Poisson’s ratio of hydrogel and Ecoflex, an electronic universal testing machine was employed to compress the prepared cylindrical hydrogel and Ecoflex to 20% strain, and then the transverse strain was measured by Vernier caliper. Finally, Poisson’s ratio was calculated by the transverse strain and the strain in the compression direction.

  1. Please add the nomenclature with appropriate units for variables used in equations.

Reply to the comment:

Thanks for the valuable suggestion. We have made corresponding additions in the revised manuscript as follows:

In the above Equations, the unit of force, length and resistance are unified as Newton (N), millimeter (mm) and Ohm (Ω), respectively.

  1. The details concerning FEM simulation have to be presented.

Reply to the comment:

Thanks for the suggestion. The details content of FEM simulation was included in the supporting information as follows:

The standard/explicit module of Abaqus was used to build and analyze the model. The overall size of the hydrogel and Ecoflex layer in the model were consistent with the actual size of the encapsulated hydrogel. Figures S8a and S8b are the model images after “view cut” operation from the X-plane and Z-Plane planes, respectively. It can be seen that the hydrogel was completely encapsulated in the Ecoflex, and the electrodes were placed on the upper and lower surfaces of the hydrogel. For the material property setting, the Young's modulus of the hydrogel, Ecoflex, and nickel-chromium wire were set as 0.012 MPa, 0.2 MPa, and 2240 GPa, respectively, their Poisson's ratios were set as 0.49, 0.45, and 0.3, respectively. The Poisson’s ratio of nickel chromium wire is obtained by querying the manufacturer. In order to obtain the Poisson’s ratio of hydrogel and Ecoflex, we used an electronic universal testing machine to compress the prepared cylindrical hydrogel and Ecoflex to 20% strain, and then measured the transverse strain by Vernier caliper. Finally, Poisson’s ratio is calculated by the transverse strain and the strain in the compression direction. For the boundary condition setting, as shown in Figure S9d, a boundary condition (U1= U2= U3= UR1= UR2= UR3= 0) was created on the lower surface of the model. A vertical load was applied to the upper surface of the model, resulting in a 20% strain in the model. Eight-node linear hexahedral element (C3D8) was selected as the three-dimensional finite element model of hydrogel and Ecoflex; Two-node linear three-dimensional truss element (T3D2) was selected as the three-dimensional finite element model of nickel-chromium wire. The approximate global size was set to 0.5, and the maximum deviation factor was 0.1. The meshing result is shown in Figure S8c and 8e.

Figure S8. (a) The image of the model after “view cut” operation from the X-plane; (b) The image of the model after “view cut” operation from the Z- plane; (c) Schematic diagram of meshing of electrodes; (d) Schematic diagram of boundary conditions; (e) Schematic diagram of meshing of hydrogel and Ecoflex

  1. The FEM simulations should be compared with experiments.

Reply to the comment:

Thank you for your valuable advice. In the experiment, when the compression strain is set at 20%, the top of the sensor will be shifted down by 3.6 mm, which was highly consistent with the simulation results as indicated Figure 8a and Figure S9.

  1. There are multiple mistakes in the References (e.g. “et al.”)

Reply to the comment:

Thank you for detailed comment. We are truly sorry for the mistakes, and have carefully checked and revised in the revised manuscript.
